# The *ctpF* Gene Encoding a Calcium P-Type ATPase of the Plasma Membrane Contributes to Full Virulence of *Mycobacterium tuberculosis*

**DOI:** 10.3390/ijms23116015

**Published:** 2022-05-27

**Authors:** Milena Maya-Hoyos, Dulce Mata-Espinosa, Manuel O. López-Torres, Blanca Tovar-Vázquez, Jorge Barrios-Payán, Juan C. León-Contreras, Marisol Ocampo, Rogelio Hernández-Pando, Carlos Y. Soto

**Affiliations:** 1Chemistry Department, Faculty of Sciences, Universidad Nacional de Colombia, Ciudad Universitaria, Carrera 30 N° 45-03, Bogota 111321, Colombia; mmayah@unal.edu.co; 2Department of Pathology, Experimental Pathology Section, National Institute of Medical Sciences and Nutrition ‘‘Salvador Zubirán”, Mexico City 14080, Mexico; dulmat@yahoo.com.mx (D.M.-E.); lopeztorresmanuel88@gmail.com (M.O.L.-T.); blanka.tovaz@gmail.com (B.T.-V.); qcjbp77@yahoo.com.mx (J.B.-P.); jcleonc@hotmail.com (J.C.L.-C.); 3Fundación Instituto de Inmunología de Colombia (FIDIC), Carrera 50 # 26-20, Bogota 111321, Colombia; marisol.ocampo26@gmail.com

**Keywords:** tuberculosis, *Mycobacterium tuberculosis*, P-type ATPases, *ctpF*, calcium, virulence, attenuated strain

## Abstract

Identification of alternative attenuation targets of *Mycobacterium tuberculosis* (*Mtb*) is pivotal for designing new candidates for live attenuated anti-tuberculosis (TB) vaccines. In this context, the CtpF P-type ATPase of *Mtb* is an interesting target; specifically, this plasma membrane enzyme is involved in calcium transporting and response to oxidative stress. We found that a mutant of *Mtb*H37Rv lacking *ctpF* expression (*Mtb*Δ*ctpF*) displayed impaired proliferation in mouse alveolar macrophages (MH-S) during in vitro infection. Further, the levels of tumor necrosis factor and interferon-gamma in MH-S cells infected with *Mtb*Δ*ctpF* were similar to those of cells infected with the parental strain, suggesting preservation of the immunogenic capacity. In addition, BALB/c mice infected with *Mtb*∆*ctpF* showed median survival times of 84 days, while mice infected with *Mtb*H37Rv survived 59 days, suggesting reduced virulence of the mutant strain. Interestingly, the expression levels of *ctpF* in a mouse model of latent TB were significantly higher than in a mouse model of progressive TB, indicating that *ctpF* is involved in *Mtb* persistence in the dormancy state. Finally, the possibility of complementary mechanisms that counteract deficiencies in Ca^2+^ transport mediated by P-type ATPases is suggested. Altogether, our results demonstrate that CtpF could be a potential target for *Mtb* attenuation.

## 1. Introduction

Tuberculosis (TB) is the leading cause of death in humans by a single bacterial pathogen (1.3 million deaths in 2020), which is now affected by the COVID-19 pandemic [1]. The emergence of multidrug and extensively drug-resistant (MDR and XDR) *Mycobacterium tuberculosis* (*Mtb*) strains and co-infection of *Mtb* with HIV have made TB a serious threat to global health [1,2].

The Bacillus Calmette–Guérin (BCG), originally an isolate of *M. bovis*, is the only approved vaccine against TB [2]. BCG is effective against disseminated forms of TB in pediatric populations but shows variable protection levels against pulmonary disease in adolescents and adults [3,4,5]. The irregular protection conferred by BCG has been attributed to various factors, including the heterogeneity of derived strains and the absence of immunodominant antigens, such as the early secretory antigen 6 kDa (ESAT-6) and culture filtrate protein 10 (CFP-10); this is a consequence of the RD1 gene deletion, which in turn diminishes the induction of long-term memory responses against *Mtb* infection [2,4,6]. Therefore, to significantly decrease the high incidence of TB (9.9 million new cases in 2020) [1], there is an urgent need to develop new effective vaccine candidates against respiratory forms of TB [2].

Currently, there are several TB vaccine candidates under development, which aim to replace BCG (the prime vaccine) or to be a booster for neonatal BCG vaccination to enhance pre-existing immunity and compensate for the long-term protection deficiencies of BCG [2,5,7]. Among these candidates, live attenuated *Mtb* strains are potential anti-TB vaccines because they display a broad antigenic repertoire and maintain the whole T cell epitopes that elicit an efficient protective immunity [3,5]. Attenuated *Mtb* strains mimic natural TB infection without causing significant pathological alterations or disease [2,8], and confer natural immunity against reinfection or concomitant secondary infections [4,9]. 

The design of attenuated strains of *Mtb* requires the identification of key targets for attenuation to ensure safety and stable genetic deletions [8,10]. The transmembrane proteins that aid in maintaining ionic homeostasis and generating the appropriate electrochemical gradients for cell survival constitute a group of potential target proteins for bacterial attenuation [11,12,13,14]. Transport systems related to transmembrane proteins are critical for *Mtb* survival, as they preserve ion concentrations at the suitable nutrient level for proper cellular function [15,16]. Excessive accumulation of cations, including Ca^2+^, is toxic and can block functional groups, displace essential ions and modify active conformations of bacterial biomolecules [17]. 

Although bacteria do not carry out some of the cellular processes of eukaryotic cells, there is evidence that some of these processes are regulated by changes in the cytosolic concentration of free Ca^2+^ [18], such as growth [19], motility [20], quorum sensing [21], sporulation [22] and the development of different bacterial structures [23]. Cellular processes involving calcium transport in *Mtb* remain mostly unexplored. 

The intracellular concentration of Ca^2+^ in bacteria, including *Mtb*, is strictly regulated and maintained at very low levels (100 nM approximately) [24], while the external Ca^2+^ concentration reaches up to 300 mM [25]. Bacteria spend energy to activate specialized transport systems across the plasma membrane in order to maintain tight Ca^2+^ concentration gradients [26]. Bacteria possess calcium-binding proteins (CaBPs), which, together with Ca^2+^/H^+^ type antiporters and calcium channels, help to attenuate the intracellular concentration of Ca^2+^ and to maintain ion homeostasis across the plasma membrane [26,27]. CaBPs fix cytosolic Ca^2+^ and help to attenuate the toxicity produced by high metal concentrations. A common characteristic of CaBPs is the presence of calcium-binding motifs of the EF-hand or Greek-key type [28].

Concerning phagocytosed mycobacteria, *Mtb* contributes to lower Ca^2+^ concentration inside the phagosome, therefore preventing phagolysosome fusion and allowing bacterial proliferation [29]. In this case, the external adhesion protein, PE_PEGR, which contains the glycine-rich motif GGXGXD/NXUX, binds extracellular Ca^2+^, diminishing the Ca^2+^ concentration in the phagosome and thus its maturation [30].

Moreover, transport systems such as P-type ATPases also play an important role in the ion homeostasis of the mycobacterial cell membrane [15,16], and respond to toxicity produced by high levels of metals within human macrophages during infection, so their deletion disturbs these detoxification systems and reduces the virulence of the bacillus [31,32,33]. 

Membrane proteins transport different cations (e.g., Na^+^, K^+^, H^+^, Cu^+^, Cu^2+^, Ca^2+^, Mn^2+^, Co^2+^, Cd^2+^, Zn^2+^), and phospholipids across cell membranes use the energy released by the hydrolysis of ATP [34,35]. Twelve P-type ATPases have been identified in the *Mtb* genome and classified according to ion specificity and transmembrane topology: seven P1B-type (heavy metal pumps; CtpA, CtpB, CtpC, CtpD, CtpG, CtpJ and CtpV), four P2-type (encoding an alkali/alkaline earth metal transporter; CtpE, CtpF, CtpH and CtpI) and one P1A-type ATPases (potassium transporter; KdpB) [35,36].

Regarding the functional characterization of P2-type ATPases, it is worth noting that CtpE is responsible for Ca^2+^ uptake in *M. smegmatis* [37]; CtpE is specifically associated with the acquisition of extracellular calcium. The *ctpE* gene is part of an operon negatively regulated by calcium concentration. Therefore, *ctpE* is activated at low Ca^2+^ concentrations and this allows mycobacteria to grow at low Ca^2+^ levels. Apparently, Ca^2+^ imported by CtpE is key for the integrity of the *M. smegmatis* cell wall [37]. On the other hand, CtpH is involved in calcium pumping in mycobacterial cells, and it is possibly also related to mycobacterial survival under toxic Ca^2+^ concentrations [38]. A characterization of ions regulated by CtpI transport across cell membranes has not yet been performed.

Among the 12 existing P-types of ATPases identified in the *Mtb* genome [35,36], the *ctpF* gene (encoding an alkaline/alkaline earth metal transporter) shows the highest level of activation under different conditions, including oxidative/nitrosative stress, hypoxia and infection [39]. It had been previously established that CtpF is a membrane transporter involved in calcium pumping outside mycobacterial cells, and in the response to oxidative/nitrosative stress [40]. Moreover, *ctpF* is the only *Mtb* P-type ATPase gene regulated by the global latency regulator (DosR), which is involved in the adaptation of bacilli to anaerobic environments, acidic pH, starvation and high nitric oxide (NO) intermediates inside macrophages and granulomas; thus, *ctpF* promotes a non-replicating persistence (NRP) or dormancy state [41,42,43,44,45]. Therefore, the fact that *ctpF* is part of the DosR regulon and its expression is activated in response to redox stress (a condition faced by the tubercle bacillus in the phagosomal environment) suggests that the role of this transporter is critical during *Mtb* infection. Interestingly, a previous study established that calcium efflux mediated by *Mtb* CtpF inhibits mTOR-dependent autophagy and enhances bacterial survival [46]. 

In this work, we observed that the *ctpF* deletion alters mechanisms that allow *Mtb* to survive and multiply within mouse alveolar macrophages (MH-S) leading to reduced virulence in a mouse model of pulmonary TB. In addition, we analyzed the expression levels of *ctpF* in a mouse model of latent and progressive TB, revealing that its expression favors *Mtb* persistence under unfavorable conditions such as the dormancy state. Finally, a complementary alternative to counteract deficiencies in the ion transport produced by the *ctpF* deletion is discussed.

## 2. Results

### 2.1. The ctpF Deletion Does Not Alter the Mtb Growth Kinetics in Standard Culture 

Before comparing the intracellular proliferation of *Mtb*H37Rv and *Mtb*Δ*ctpF* strains in MH-S cells, the kinetics of growth of both strains in standard cultures was assessed. As observed in Figure 1A, the mutant (*Mtb*Δ*ctpF*) and the parental (*Mtb*H37Rv) strains reached the mid-logarithmic phase of growth at 15–16 days (OD_600_ ≈ 1.11–1.36), and 16–17 days (OD_600_ ≈ 1.08–1.27), respectively. Furthermore, the mutant strain exhibited a higher growth rate in the logarithmic phase (0.220 OD_600_/day), compared to the parental strain (0.192 OD_600_/day) (Figure 1A). In addition, the mutant strain did not show morphological differences relative to the parental strain (rough cream-colored colonies) (Figure 1B). However, in contrast to the parental strain, the *Mtb*Δ*ctpF* strain was unable to fix neutral red (RN) and did not show appreciable differences to the coloration showed by the attenuated strain *Mtb*H37Ra, suggesting there were unknown changes at the level of cell wall composition in the mutant strain (Figure 1C).

Figure 1 shows that *ctpF* is not essential for the growth of *Mtb* in culture under standard conditions, suggesting that the *ctpF* gene was not required for optimal growth of *Mtb* in vitro [47]. On the other hand, the NR test indicates that the *ctpF* mutation alters the ability of *Mtb*H37Rv to bind and reduce the NR colorant, suggesting modifications of the cell wall composition in the mutant strain (Figure 1C) and possibly an impaired virulence [48]. Transmission Electron Microscopy (TEM) analysis of the bacterial ultrastructural morphology of *Mtb*Δ*ctpF* showed anomalies relative to the parent *Mtb*H37Rv strain (Figure 2A–D). The *Mtb*Δ*ctpF* strain showed an irregular shape with projections and concavities of the cell wall, which was thinner than that of the parent *Mtb*H37Rv strain; both the external electron-dense and the inner peptidoglycan layers were thinner in the mutant strain (Figure 2A–D).

To verify whether the absence of an ATPase involved in the Ca^2+^ efflux interferes with the virulence of *Mtb*, as suggested by potential alterations in the cell wall composition, in vitro and in vivo assays of bacterial infection were performed.

### 2.2. ctpF Is Required for Mtb Intracellular Proliferation in MH-S Cells

To evaluate whether the *ctpF* gene is relevant for the intracellular growth of *Mtb*, an infection model of MH-S cells was used. Both parental and mutant strains showed similar levels of infection of MH-S cells at 1 h (the time at which mycobacteria are phagocytosed [49]), indicating that the *ctpF* deletion does not alter the initial ability of *Mtb* to infect phagocytic cells, such as MH-S cells (Figure 3A). However, when the infection process advanced, a significant decrease in replication of the mutant strain, relative to the parental strain, was observed, especially between 3 and 7 dpi (**** *p* < 0.0001) (Figure 3A). Indeed, the *Mtb*H37Rv strain showed a 112-fold increase in the number of colony-forming units (CFU) during 7 dpi (1 h and 7 dpi: 5033 and 566,667 CFU/mL, respectively), while the mutant *Mtb*Δ*ctpF* strain showed only a 6-fold increase (1 h and 7 dpi: 4880 and 27,333 CFU/mL, respectively). Moreover, our ultrastructural study of infected MHS cells showed more phagosomes, phagolysosomes and autophagosomes in macrophages infected with the mutant strain than in those infected with the parental strain (Figure 2E–G), which indicates that the *Mtb*∆*ctpF* strain was eliminated more easily. These results suggest that CtpF is required for the survival and optimal intracellular multiplication of *Mtb* in MH-S macrophages (Figure 2 and Figure 3).

### 2.3. Deletion of ctpF Does Not Significantly Affect the Production of IFN-γ and TNF in Infected MH-S Cells

It is known that live-attenuated or heat-inactivated mycobacteria maintain some immunostimulatory properties due to their ability to induce the expression of innate immunity cytokines [50]. Therefore, our next step was to assess whether the *ctpF* deletion could affect the immunogenicity of *Mtb*. Thus, the level of interleukin-12 (IL-12), IFN-γ and TNF produced by macrophages infected with parental and mutant strains was quantified (Figure 3B). The *Mtb*Δ*ctpF* strain stimulated the production of TNF and IFN-γ at levels that were similar to those induced by the *Mtb*H37Rv strain (Figure 3B). On the other hand, the levels of IL-12 were very low, and it was not possible to quantify them using this methodology. The production of TNF in MH-S cells infected with the mutant strain was significantly lower (113 pg/mL) at 7 dpi compared to cells infected with the parental strain (150 pg/mL) (* *p* < 0.05; Figure 3B). This could be linked to lower bacterial load and lower stimulation by mycobacterial antigens of the mutant strain in the phagocytic cells (Figure 3A). The *Mtb*Δ*ctpF* strain exhibited lower intracellular proliferation compared to the *Mtb*H37Rv strain (21-fold CFU reduction: 27,333 vs. 566,667 CFU/mL) at 7 dpi (Figure 3A).

IFN-γ levels were measured at different time periods post-infection, but they were only detected in MH-S cells infected at 7 dpi or at the end of the in vitro infection assays (Figure 3B), confirming that macrophages can express this cytokine in response to mycobacterial stimuli, such as structural components (e.g., lipopolysaccharides) and/or cytokines (stimulating feedback), at infection stages close to the onset of the adaptive immune response (starting at week 2 post-infection) [51,52,53]. Thus, MH-S macrophages infected with the *Mtb*H37Rv and *Mtb*Δ*ctpF* strains are stimulated similarly, indicating that in vitro assays the *ctpF* deletion does not significantly affect the immunogenicity of *Mtb* while inducing the production of two key cytokines (TNF and IFN-γ) required for protection against the *Mtb* infection [54]. 

### 2.4. The MtbΔctpF Strain Shows Attenuated Virulence in Mice

To obtain information on the importance of CtpF for the virulence of *Mtb* in vivo, a survival assay using BALB/c mice (five animals per strain) inoculated intratracheally with 2.5 × 10^5^ CFU of the *Mtb* strains was performed. Mice infected with the *Mtb*H37Rv strain showed 100% mortality at 84 days of infection, while at this time 40% of mice infected with the *Mtb*Δ*ctpF* strain were still alive (Figure 4A). The median survival time of *Mtb*H37Rv-infected mice was 59 days, in contrast to 84 days for the *Mtb*Δ*ctpF*-infected mice (Figure 4A). Indeed, animals infected with *Mtb*Δ*ctpF* died at 112 dpi, showing a 28-day increase in life span. During survival assays, some signs of the disease (such as weight loss and anorexia) were also measured. Unlike mice infected with the *Mtb*Δ*ctpF* strain, which had a lower average weight from the tenth week post-infection, mice infected with the *Mtb*H37Rv strain began to lose weight in a sustained manner from the sixth week post-infection (Figure 4B). Mice infected with the parental strain *Mtb*H37Rv lost significantly more body weight, compared to animals infected with the *Mtb*Δ*ctpF* strain, throughout the infection assays (Figure 4C).

### 2.5. Other Genes Encoding P2-Type ATPases Are Activated in MtbΔctpF Strain under Stress Conditions

To determine whether any of the P2-type ATPases (alkaline/alkaline earth metal transporters) genes might be upregulated in the absence of *ctpF* under different stress conditions, the transcriptional pattern of the genes *ctpF*, *ctpH*, *ctpE* and *ctpI* was analyzed in the *Mtb*Δ*ctpF* strain. An unrelated *ctpA* gene that codes a Cu^+^-ATPase [55] was included as a control. As shown in Figure 5A, the mRNA level of *ctpH* increased 180-fold in the *Mtb*Δ*ctpF* strain after exposure to sublethal doses of Ca^2+^ (half of the maximal inhibitory concentration (IC_50_) ≈ 2.5 mM [40]) for 3 h, relative to control cells (*Mtb*Δ*ctpF* cells that were not intoxicated; transcription ratio ≈ 1.00). Conversely, the transcription levels of the *ctpE* and *ctpI* genes were close to those of the control gene (*ctpA*) and were not significantly different from control cells (unstressed bacteria) (Figure 5A).

Additionally, we evaluated whether the mutant strain activated any P2-type ATPase gene during in vitro infection as a complementary mechanism produced by the impairment of the metal transport mediated by CtpF in the *Mtb*Δ*ctpF* strain. For this purpose, we compared the mRNA levels of P2-type ATPases (*ctpF*, *ctpH*, *ctpE* and *ctpI*) and *ctpA* genes in strains *Mtb*Δ*ctpF* and *Mtb*H37Rv (control strain; transcription ratio ≈ 1.00) during the infection of MH-S cells. Figure 5B shows that the ratios between the mRNA levels of *ctpI* and *cptA* in the mutant and parental strains at different dpi were close to 1 (Figure 5B). However, the transcription of *ctpH* and *ctpE* genes increased 2- to 5-fold in *Mtb*Δ*ctpF*, relative to the *Mtb*H37Rv strain, throughout the infection (from 1 to 7 dpi). 

### 2.6. ctpF Transcription Was Higher during Experimental Latent TB Infection

As stated above, the expression of *ctpF* is controlled by DosR [44], which is involved in maintaining low bacterial growth under unfavorable conditions (such as in granulomas), entering into a dormant state and maintaining it for a long time within the host [41,42,43,44,45,56]. Therefore, we used qRT-PCR to compare the absolute transcription levels of *ctpF* in a model of progressive pulmonary TB and latent TB infection in BALB/c mice (Figure 6).

The absolute expression of *ctpF* in the active TB model, evaluated at 21 and 60 dpi, was not significantly different (*p* = 0.9976), indicating that the expression of this transporter is similar in early (21 dpi, 668 copies/µL) or chronic (60 dpi, 753 copies/µL) TB (Figure 6). Conversely, the quantity of *ctpF* transcript increased as the latent infection progressed. Indeed, the *ctpF* copy number was 8359 copies/µL at 5 months post-infection (mpi) and 13,748 copies/µL at 7 mpi (**** *p* < 0.0001). Interestingly, absolute expression of *ctpF* during the latent infection TB was significantly higher than during the progressive infection TB (≈11- to 18-fold *ctpF* gene copies/µL in the latent infection relative to the progressive infection), suggesting that CtpF plays a role in the dormant state of the bacteria and could be required for persistence in vivo (Figure 6).

## 3. Discussion

When *Mtb* is phagocytosed by macrophages, tubercle bacilli then face an approximately 4-fold increase of the intraphagosomal Ca^2+^ concentration at 24 h post-infection [15,57]. Cells respond to different environmental stimuli by transient changes in the cytosolic Ca^2+^ concentration, which are useful to transmit information and to trigger cellular events according to the speed, timing and magnitude of the metal signal [26,58]. Some bacteria detect and respond to different environmental stimuli using two-component systems and Ca^2+^ sensors, which transduce signals through phosphorylation events and protein–protein interactions and thereby regulate transcription and translation processes [27,59].

However, Ca^2+^ homeostasis requires an orchestrated activity, in which active transport systems (primary and secondary), CaBPs proteins and cytosolic Ca^2+^ stores (calcisome acid and polyphosphate granules) work together in order to keep intracellular Ca^2+^ at the nanomolar range (100–300 nM) and to prevent toxicity by high metal concentrations [26,27,37]. *Mtb* also activates detoxification systems through P-type ATPases to maintain cellular homeostasis of monovalent and divalent cations, including Ca^2+^, and generate appropriate electrochemical gradients [15,16].

In a previous study, we demonstrated that the CtpF a Ca^2+^-ATPase is activated in response to redox stress, a condition faced by *Mtb* during infection [40]. This motivated us to evaluate the importance of CtpF during infection in vivo and in vitro. We suggest that CtpF could play a role in counteracting the toxic effects of the intraphagosomal environment. This behavior is in keeping with a recent study showing that knocking down *ctpF* in *Mtb*H37Rv (by CRISPR-cas9) leads to decreased mycobacterial survival in THP-1 macrophages. Specifically, at 3 dpi, the parental strain showed a 1.26 log increase from their initial CFU, while the mutant strain did not show any increase in the number of CFU at the same time of infection [46]. Thus, both studies suggest that CtpF (an ATPase involved in restoring Ca^2+^ homeostasis [40]) is necessary for *Mtb* proliferation and survival in the host cells.

The cell envelope is a cellular structure of *Mtb* that contributes to bacterial growth and survival under stressful conditions [60]. Indeed, the cell envelope is essential for *Mtb* survival because it is both a barrier for antibiotics and a modulator of the host immune response [61,62]. *ctpF* mutation had an effect on bacterial morphology and cell wall thickness, which may contribute to a decreased ability to adapt and survive in the host cell. This observation is in keeping with the negative NR test of the mutant *Mtb*∆*ctpF* strain. The fact that *Mtb* does react against NR in an alkaline environment means that mutant cells are deficient in more than one type of methyl-branched lipid in the cell wall [48]. Deficiencies in this kind of lipids are associated with decreased virulence of the tubercle bacillus [48]. It would be very interesting to identify the lipids of the *Mtb*∆*ctpF* cell wall that are absent or altered and contribute to changes in the structure of the cell wall (evidenced by TEM), and to impaired *Mtb* virulence.

TNF and IFN-γ act synergistically to induce the production of reactive oxygen and nitrogen species (RONS) through nitric oxide synthase (iNOS) [63]. The fact that MH-S cells efficiently controlled the *Mtb*Δ*ctpF* strain might be partially explained by the hypersensitivity of this strain to RONS [40]. This was supported by our ultrastructural study of infected macrophages, which showed more phagosomes and phagolysosomes in cells infected with the mutant *Mtb*∆*ctpF* than with the parent strain *Mtb*H37Rv, indicating more efficient bacteria killing in macrophages infected with the mutant strain.

On the other hand, Ca^2+^ is involved in important biological processes of prokaryotes such as chemotaxis, cell division, heterocyst differentiation, sporulation, biofilm formation, motility, genetic regulation, enzymatic activity and cell wall integrity, among others [26,27]. Furthermore, several intracellular pathogens respond to the high Ca^2+^ concentration inside host cells, triggering molecular processes that alter the host’s defense mechanisms and produce persistent infections [59,64]. 

In addition, Ca^2+^ efflux orchestrated by *Mtb* may modulate signaling events in the host, such as inhibition of autophagy and impaired phagosome maturation [46,64,65]. A previous study showed that *Mtb* inhibits mTOR-dependent autophagy and enhances bacterial survival by using efflux systems such as CtpF to pump Ca^2+^ into macrophages during the early stages of infection (1–4 h) [46]. This finding reinforces the idea that the Ca^2+^-ATPase CtpF is relevant for the regulation of molecular processes that favor the survival and virulence of *Mtb*.

The interaction of *Mtb* with host cells involves diverse mechanisms, including ion homeostasis [46]. Therefore, the ability to detect metals in the medium and modulate ion homeostasis may be a key factor for successful infection [59,66]. The fact that *Mtb* displays a differential expression of P-type ATPases during infection processes indicates that the phagosomal ion concentration changes the activity of mycobacterial transporters [15,39,46,57]. Thus, the fact that *Mtb*Δ*ctpF* strain activates *ctpH* (2–5-fold) and *ctpE* (2–5-fold) at different time periods post-infection would strongly suggest that tubercle bacillus uses several P2-type ATPases to maintain a balanced ion environment in response to stressful conditions within the host.

Since the mutant strain is more sensitive to oxidative and nitrosative stress [40], it may induce Ca^2+^ uptake by activating transporters, such as CtpE, to draw Ca^2+^ from the intracellular host stores [37,46] as a molecular mechanism to cope with oxidative stress. Certainly, a correct intracellular Ca^2+^ concentration positively modulates the expression of antioxidant proteins in defense against redox stress [26,27,67]. However, the increase in Ca^2+^ levels should be transient to maintain bacterial viability and avoid toxicity [40,46]. Thus, the mutant *Mtb*Δ*ctpF* strain may positively regulate the expression of the *ctpH* gene to compensate for the *ctpF* deficiency and restore the physiological levels of Ca^2+^ for intracellular survival.

Finally, we evaluated the absolute expression of the *ctpF* gene in a model of progressive and latent TB infection in BALB/c mice. In the case of the progressive pulmonary TB model, we evaluated *ctpF* transcript levels at 21 and 60 dpi, as in this model 21 dpi (acute or early phase) is the time of maximal protective activity mediated by IFN-γ, IL-12, TNF and NO production [68,69]. At 60 dpi (late or chronic phase) the activity of Th1 lymphocytes decreases and Th2 lymphocytes emerge, which produce anti-inflammatory cytokines, favoring bacterial survival and proliferation [68,69]. Some studies have reported that the DosR regulon (a set of at least 48 genes) contains various putative effector proteins with a vital role in maintaining the dormant state in *Mtb* [41,45]. The fact that *ctpF*, which is part of the DosR regulon, is regulated in latent infection (≈2-fold increase between 5 and 7 mpi) would indicate that CtpF could be important for the adaptation of *Mtb* to the host environment and persistent infection. 

Furthermore, it has been established that the DosR regulon is required for *Mtb* to establish an NRP state and survive during hypoxia [42], which supports the idea that CtpF is an interesting attenuation target to design novel vaccines, and may even be a new target for anti-TB compounds [38,70].

Therefore, we consider CtpF to be a potential attenuation target, since this P-type ATPase contributes to the resistance to toxic concentrations of Ca^2+^ [40] and *ctpF* deletion does not alter the immunogenicity of the mycobacterial strain, mimicking a natural infection and inducing the production of essential cytokines for the control of the bacillus [54]. However, the results obtained in this work ought to be completed with further studies using an unmarked double mutant deleting *ctpF* and other non-related genes or virulence factors, which could be considered candidates for a live vaccine used for BCG replacement.

## 4. Materials and Methods

### 4.1. Bacterial Strains and Growth Conditions

The bacterial strains and primers used in this work are listed in Appendix A. The isogenic *ctpF* mutant of *Mtb*H37Rv (*Mtb*Δ*ctpF*) was obtained as previously described [40] (Appendix A). *Mtb* strains were cultured in Middlebrook 7H9 broth (271310 BD) supplemented with 10% (*v*/*v*) OADC (Catalog 211886, BD Biosciences, Heidelberg, Germany), 0.2% glycerol (G5516, Sigma-Aldrich, St. Louis, MO, USA) and 0.05% tyloxapol (T8761, Sigma-Aldrich, St. Louis, MO, USA), at 37 °C and 5% CO_2_ with gentle agitation (80 rpm) until the mid-logarithmic phase was reached (OD_600_ ≈ 1.0–1.3), or on Middlebrook 7H10 and 7H11 agar (Catalog 262710 and 212203, respectively, BD Biosciences, Heidelberg, Germany) supplemented with OADC and 0.5% glycerol.

Bacteria cultured in 7H9-OADC were harvested by centrifugation and stored in aliquots of 1 mL with 20% glycerol at −80 °C. When required, 7H9, 7H10 and 7H11 media were supplemented with 20 μg/mL kanamycin (Kan) and 100 μg/mL hygromycin (Hyg).

Bacterial viability was checked before use. For the kinetics of mycobacterial growth, isolated colonies of *Mtb*H37Rv and *Mtb*Δ*ctpF* strains were inoculated in 60 mL 7H9-OADC, and cultured at 37 °C, 5% CO_2_, and 80 rpm for 1 week. Then, the OD_600_ of 1 mL aliquots of mycobacterial cultures was measured every day in a spectrophotometer (Thermo Fisher Scientific, Waltham, MA, USA) until the stationary phase was reached (OD_600_ ≈ 2.0–2.5).

### 4.2. NR Staining

The NR staining was performed as described previously [48,71], with some modifications. Briefly, *Mtb* strains were grown on 7H9-OADC medium at 37 °C and 5% CO_2_ with gentle agitation (80 rpm) until reaching OD_600_ ≈ 1.0–1.3. Bacterial cells were collected by centrifugation at 5500 rpm for 10 min and the cell pellet was resuspended in 5 mL of 50% aqueous methanol, then transferred to screw-cap glass tubes and incubated at 37 °C for 1 h. Supernatants were discarded, and the cells were washed with 5 mL 50% aqueous methanol and incubated at 37 °C for 1 h. Then, the supernatant was removed, and the pellets were resuspended in 5 mL 0.002% NR in Tris-HCl buffer (0.1 M Tris-HCl, pH 9.8), and incubated at room temperature for 24 h. Finally, cell staining was observed and compared to controls. Red staining of mycobacteria is considered a positive reaction (RN+), while yellow staining is rated as a negative reaction (RN−) [48].

### 4.3. MH-S Cells Culture

The MH-S cell-line (ATCC^®^ CRL-2019) was cultured to near 80–90% confluence in RPMI-1640 (with L-glutamine, 25 mM HEPES, 0.2% NaHCO_3_, pH 7.4, R6504 Sigma) supplemented with 10% fetal bovine serum (FBS) (Catalog 26140-079, Gibco Life technologies, Grand Island, NY, USA) at 37 °C and 5% CO_2_ atmosphere.

### 4.4. Infection of MH-S Cells with Mtb Strains

MH-S cells were seeded in RPMI-10% FBS medium onto 12-well culture dishes (SC-204444, Ultra Cruz, Santa Cruz Biotechnology, Dallas, TX, USA) at 5 × 10^4^ cells per well and grown at 37 °C and 5% CO_2_ for 24 h. Then, the medium was removed and 500 µL RPMI containing 2 × 10^5^ CFU of *Mtb* strains was added to each well (three wells per strain) to obtain a MOI of 1:2. Before infection, the suspended bacteria were sonicated for 30 s at 20 kHz. The plates were incubated at 37 °C and 5% CO_2_ for 1 h to allow macrophages to phagocytose the bacteria. After in vitro infection the remainder of the bacterial inoculum was plated onto 7H10-OADC, to confirm the number of CFU used. Then, the RPMI medium was removed, and cells were washed 3 times with 1 mL RPMI supplemented with 2% antibiotic solution (10,000 U/mL penicillin and 10,000 µg/mL streptomycin, Catalog 10378-016, Gibco, Grand Island, NY, USA) to remove extracellular bacteria. Then, cells were carefully resuspended in 1 mL RPMI-10% SFB medium and incubated at 37 °C and 5% CO_2_ during different incubation times (1 h, and 1, 3, and 7 days).

After each incubation period, supernatants of cell cultures were recovered (for cytokine detections) and immediately frozen at −80 °C until use. MH-S cells were lysed with 200 μL 0.1% SDS-7H9 for 10 min at room temperature; lysis was stopped by adding 200 μL 20% bovine serum albumin (BSA)-7H9. Subsequently, serial dilutions of lysates were prepared in 7H9-OADC medium, and cells were grown in 7H10-OADC. The viability of intracellular bacteria was determined by counting CFU after 21 days of incubation at 37 °C in 5% CO_2_.

### 4.5. Quantification of Cytokines in MH-S Cells Infected with Mtb Strains

Cytokines (IL-12, TNF and IFN-γ) were measured in culture supernatants of infected MH-S cells by sandwich ELISA using the OptEIA™ BD Biosciences kit (Heidelberg, Germany, Catalog 555165, 558534 and 555138, respectively) according to the manufacturer’s instructions. Initially, 100 µL capture antibody diluted in the capture buffer was added to 96-well plates (2580 Costar). The plates were sealed with parafilm^®^ and incubated at 4 °C overnight (>16 h). Then, the plates were washed 3 times with 300 µL wash buffer (Phosphate-buffered saline (PBS)-0.05% Tween 20) using the Wellwash™ microplate washer (Thermo Fisher Scientific, Waltham, MA, USA). The plates were blocked by adding 200 µL PBS-10% BSA and incubated at room temperature for 1 h. The plates were then washed 3 times with 300 µL wash buffer using a microplate washer. Then, 100 µL of each standard dilution (1000—15.6 pg/mL IL-12, 1000—15.6 pg/mL TNF and 4000—62.5 pg/mL IFN-γ) or culture supernatants were added to wells and incubated at room temperature for 2 h. Plates were washed 5 times with 300 µL wash buffer, and 100 µL detection solution (biotinylated secondary antibody + streptavidin-horseradish peroxidase in PBS-10% SFB) was added to the wells and incubated at room temperature for 1 h. Subsequently, plates were washed 7 times with 300 µL wash buffer, 100 µL substrate solution (tetramethylbenzidine and H_2_O_2_; Catalog 55214, BD Biosciences, Heidelberg, Germany) was added, and the plates were incubated at room temperature for 30 min in darkness. Finally, reactions were stopped by adding 50 μL 2 N H_2_SO_4_ solution, and the absorbance was measured at 450 nm and 570 nm (for correction) using a microplate spectrophotometer (Epoch™ BioTek, Agilent Technologies, Santa Clara, CA, USA).

### 4.6. Progressive Pulmonary and Latent TB Infection in BALB/c Mice

The experimental model of progressive pulmonary TB has been previously described in detail [68,69]. Briefly, the bacterial suspensions were sonicated for 30 s at 20 kHz to disaggregate bacterial clumps. Male BALB/c mice (6–8 weeks old) obtained from Mexico’s Instituto Nacional de Ciencias Médicas y Nutrición Salvador Zubirán (INCMNSZ) animal house facility were anesthetized with 100 μL of Sevofluorane at 100% in a gas chamber and infected intratracheally with 2.5 × 10^5^ CFU of either *Mtb* strain (*Mtb*H37Rv or *Mtb*Δ*ctpF;* 5 mice per strain) suspended in 100 μL of saline solution-0.02% tyloxapol. After animal infection, the remainder of the bacterial inoculum was plated onto 7H10-OADC to confirm the number of CFU administered to the animals. 

Infected mice were maintained in a vertical position until spontaneous recovery; they were kept in groups of five throughout the study in cages fitted with microisolators connected to a negative pressure system (Allentown’s IVC Systems) in an animal biosafety level III facility. Animals were weighed and monitored daily to record mortality and follow disease progression. All procedures were performed in a laminar flow cabinet in a biosafety level III facility.

To induce latent infection, the same intratracheal infectious procedure was performed, but each mouse was infected with 10 CFU of *Mtb*H37Rv (3 mice per time period post-infection). After 5 and 7 months of infection, there was no bacterial growth from lung homogenates, although qRT-PCR demonstrated the expression of bacterial 16S*rRNA* and genes related to latent infection, such as iso-citrate lyase and alpha-crystallin, which are similar features of latent infection in humans (manuscript in preparation for submission).

### 4.7. RNA Extraction and cDNA Synthesis

For RNA isolation from *Mtb*Δ*ctpF* strain exposed to sublethal doses of Ca^2+^ (previously estimated in [40]), or untreated mycobacterial cells, cells were grown on 7H9-OADC medium until OD_600_ ≈ 1.0–1.3; they were then washed three times and suspended in Sauton medium. Cells were treated with CaCl_2_ (IC_50_ ≈ 2.5 mM) and incubated at 37 °C and 80 rpm for 3 h. Cells were collected by centrifugation at 7500 rpm for 10 min at 4 °C and washed 3 times with DEPC-treated H_2_O, and the cell pellet was used for RNA isolation using the TRIzol method (Invitrogen, Carlsbad, CA, USA) [72]. RNA was resuspended in 50 µL of DEPC-treated H_2_O, quantified in a NanoDrop™ OneC (Thermo Fisher Scientific, Waltham, MA, USA), and its integrity was visualized in 2% agarose gels. To eliminate possible DNA contamination, 2 μg of RNA was treated with 4 μL of DNase I (1 U/μL, EN0521 Thermo Scientific) and 1 μL of RNase inhibitor (20 U/μL, N8080119 Thermo Fisher Scientific, Waltham, MA, USA) at 37 °C for 40 min. Then, DNase was inactivated by adding 2 µL 25 mM EDTA and incubating at 65 °C for 10 min. cDNA synthesis was prepared using 2 µg of RNA and the RevertAid First Strand cDNA Synthesis kit (K1621, Thermo Fisher Scientific) and random primers (0.2 µg/µL, Thermo Fisher Scientific). cDNA was frozen at −20 °C until use. 

For total RNA isolation from in vitro infection at different dpi, infected cells were collected and resuspended in buffer RLT Plus (1053393, QIAGEN, Hilden, Germany) with 10% *β*-mercaptoethanol to preserve RNA integrity. Then, RNA extraction was performed using the RNeasy^®^ Micro kit (74004, QIAGEN) following the manufacturer’s instructions. For total RNA extraction from BALB/c mice with progressive pulmonary TB, three mice per time point (21 and 60 dpi) were sacrificed by exsanguination. The right lungs were excised aseptically, collected in 1.5 mL cryotubes containing 1 mL RLT Plus buffer with β-mercaptoethanol, frozen immediately in liquid nitrogen, and stored at −80 °C until processing. Each sample was slowly unfrozen and homogenized in the FastPrep-24^TM^ (MP Biomedicals, Irvine, CA, USA) with zirconia and flint beads (MP Biomedicals) for three cycles of 20 s. RNA extraction was carried out using the RNeasy^®^ Mini kit (74106, QIAGEN), following the manufacturer’s instructions [73]. 

For total RNA extraction from BALB/c mice with latent TB infection, three mice per time point (5 and 7 mpi) were sacrificed by exsanguination. Briefly, lungs were frozen with liquid nitrogen, pulverized using sterile mortars and pestles, and collected in 1.5 mL cryotubes containing 1 mL RLT Plus buffer with *β*-mercaptoethanol. Then, each sample was centrifuged at 14,000 rpm at 4 °C for 5 min, the supernatant was discarded, and the pellet was kept on ice. This procedure allowed the enrichment of bacterial cells in the pellet [74]. The RNA extraction was carried out from the pellet using the Quick RNA miniprep kit (Catalog R1055, Zymo Research Corporation, Irvine, CA, USA) following the manufacturer’s recommendations.

All RNAs isolated from in vitro and in vivo infection TB were treated with DNase I (1 U/μL, EN0521 Thermo Fisher Scientific), and the quantity and quality of RNAs were evaluated using the 260/280 OD ratio in the Epoch^TM^ Microplate Spectrophotometer and in 2% agarose gels. cDNA synthesis was performed using 100 ng of RNA, random primers (0.2 µg/µL, Thermo Fisher Scientific), and the Omniscript kit (205113 QIAGEN), following the manufacturer’s instructions. Finally, all the cDNAs were frozen at −20 °C until use.

### 4.8. qRT-PCR Analysis

The transcription levels of *ctpF*, *ctpH*, *ctpE*, *ctpI* and *ctpA* genes were evaluated by relative quantification with the PfaffI method using cDNA from *Mtb*Δ*ctpF* cultured under sublethal concentrations of Ca^2+^ and from in vitro infection assays [75]. In this case, the *Mtb* 16S*rRNA* gene was used to normalize gene expression. To determine the amplification efficiency of target and reference genes, 10-fold serial dilutions of *Mtb*H37Rv genomic DNA we tested.

The transcription levels of *ctpF* from in vivo infection were determined by absolute quantification using cDNA. The *ctpF* gene was amplified from *Mtb*H37Rv DNA by PCR using the primers listed in Appendix A. A PCR fragment of 248 pb was purified using the FavorPrep^TM^ Gel/PCR purification kit (FAGCK 001, FAVORGEN, Ping-Tung, Taiwan), and quantified by spectrophotometry using a NanoDrop™ One^C^ (Thermo Fisher Scientific, Waltham, MA, USA). The *ctpF* gene copy number was calculated according to the molar mass and concentration of amplicon sequence using the following formula: number of copies = (PCR fragment concentration (ng) × 6.022 × 10^23^)/(PCR fragment length × 1 × 10^9^ × 660) [76]. Ten-fold serial dilutions of the PCR fragment were prepared to generate standard curves ranging from 3.89 × 10^2^ to 3.89 × 10^8^ copies and quantified by qPCR. Then, CT values of the samples were transformed to the target gene copy numbers according to the standard curve [77]. qPCR or qRT-PCR was carried out using the iQ^TM^ SYBR^®^ Green Supermix kit (Bio-Rad Laboratories, Hercules, CA, USA) on the CFX-96 thermocycler (Bio-Rad Laboratories, Hercules, CA, USA). Cycling conditions were as follows: initial denaturation at 95 °C for 5 min, followed by 49 cycles of 95 °C for 30 s, Tm (°C) for 15 s, and 72 °C for 15 s. The quantifications were carried out in triplicate (from two independent experiments); a negative control (without cDNA) was included in the PCR reactions.

### 4.9. TEM

Ultrastructural morphology of parental and mutant strains, as well as bacterial phagocytosis and intracellular killing by MHS cells, were evaluated by TEM. Bacilli were cultured in 7H9-OADC medium until OD_600_ ≈ 1.0–1.3. Bacteria were collected, fixed by immersion in 4% glutaraldehyde in cacodylate buffer for 4 hs, followed by exposure to osmium tetroxide fumes. The bacterial suspension was centrifuged to form a pellet that was later dehydrated with graded ethyl alcohol solutions and embedded in Epon resin (London Resin Company, Aldermaston, UK). Thin sections 70 nm to 90 nm in width were placed on copper grids, contrasted with uranium salts and Reynol’s lead citrate (Electron Microscopy Sciences, Hatfield, PA, USA), and examined with an FEI Tecnai G2 Spirit Transmission Electron Microscope (Hillsboro, OR, USA).

MH-S cells were seeded onto 75 cm^2^ culture flasks (SC-200263, Ultra Cruz, Dallas, TX, USA) at 3 × 10^6^ cells per flask and incubated for 1 h at 37 °C and 5% CO_2_. Then, the RPMI-10% FBS medium was removed, and 500 µL RPMI containing 15 × 10^6^ CFU of Mtb strains was added to each flask (MOI 1:5). The flasks were incubated at 37 °C and 5% CO_2_ for 3 days. The RPMI medium was removed, and cells were washed 3 times with 1 mL RPMI supplemented with a 2% antibiotic solution (10,000 U/mL penicillin and 10,000 µg/mL streptomycin, 10378-016 Gibco). Then, the cells were scraped, and the cell suspension was collected and centrifuged for 10 min at 1500 rpm. Cells were fixed with 2.5% glutaraldehyde in cacodylate buffer (0.15 M sodium cacodylate, pH 7.2) for 30 min. Then, cells were collected and resuspended in the same fixer solution. Cells were post-fixed with 2% OsO_4_ buffer, dehydrated in graded ethyl alcohol solutions, and embedded in low viscosity Spurr’s resin (Catalog 14300, Electron Microscopy Sciences). Thin 80–90 nm sections were placed on copper grids, contrasted with uranyl acetate and Reynol’s lead citrate. Cells were examined with FEI Tecnai G2 Spirit Transmission Electron Microscope (Hillsboro, OR, USA) at 80 kV.

### 4.10. Ethics Statement

All our protocols with animals were reviewed and approved by the Internal Committee for the Care and Use of Laboratory Animals (CICUAL) of the National Institute of Medical Sciences and Nutrition ‘‘Salvador Zubirán” in Mexico City, according to the guidelines of the Mexican National Regulations on Animal Care and Experimentation (NOM 062-ZOO-1999; permit code PAT-973-13/15-1).

## 5. Conclusions

In this work, models of MH-S cells infection were used along with a progressive pulmonary and latent TB infection in BALB/c mice with the goal of evaluating the gene expression and potential effect on virulence of deleting the *ctpF* gene from the *Mtb* genome. It was initially observed that the *ctpF* deletion does not alter either the *Mtb* growth or kinetics in standard culture, and that the *Mtb*Δ*ctpF* strain exhibits impaired multiplication in MH-S cells and reduced virulence in a mouse model, indicating that CtpF plays an important role in *Mtb* virulence. Interestingly, deletion of *ctpF* does not significantly affect either the production of IFN-γ or TNF in infected MH-S cells. In addition, *ctpF* transcription in the mice’s lungs was higher during the experimental latent TB infection than in the progressive disease, suggesting that CtpF plays a role in the dormant state of mycobacteria and could therefore be required for persistence in vivo (Figure 7). 

Furthermore, it seems that the Ca^2+^-ATPase CtpF is required to maintain ion homeostasis and regulate molecular processes that favor intracellular survival within host cells. However, other genes encoding P2-type ATPases are activated in the *Mtb*Δ*ctpF* strain under stress conditions, which suggests that there exists a complementary mechanism carried out by other alkali/alkaline earth metal transporters to cope with deficiencies in calcium transport in the *Mtb*Δ*ctpF* mutant strain.

To conclude, CtpF could be a potential target for attenuation, since its deletion affects some mechanisms used by tubercle bacillus to counteract adverse conditions during the infection process. 

## Figures and Tables

**Figure 1 ijms-23-06015-f001:**
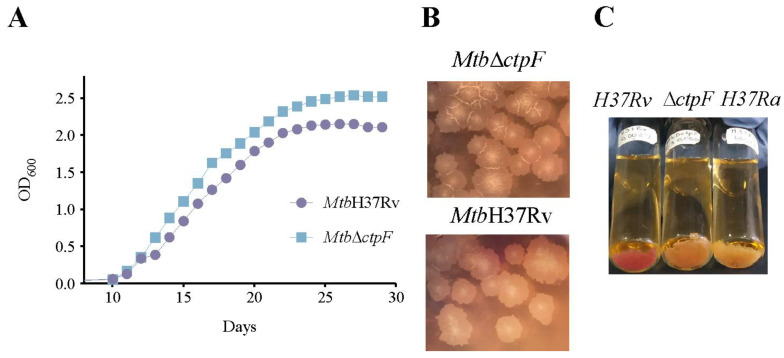
Kinetics of Growth of *Mtb*H37Rv and *Mtb*Δ*ctpF* strains in standard cultures. (**A**) The *Mtb* strains were grown in 7H9-OADC (oleic acid-albumin-dextrose-catalase)-0.05% tyloxapol medium at 37 °C with shaking (80 rpm) in 5% CO_2_. OD_600_ was monitored until reaching the stationary phase. Curves correspond to mean ± standard deviation of three technical replicates. (**B**) The strains were cultured in 7H10-OADC medium at 37 °C in 5% CO_2_ until the appearance of colonies, which exhibited rough colony morphology. (**C**) The red coloration of the *Mtb*H37Rv (virulent) strain is considered a positive reaction (RN+), while the yellow coloration of the *Mtb*H37Ra (avirulent) strain is considered a negative reaction (RN−).

**Figure 2 ijms-23-06015-f002:**
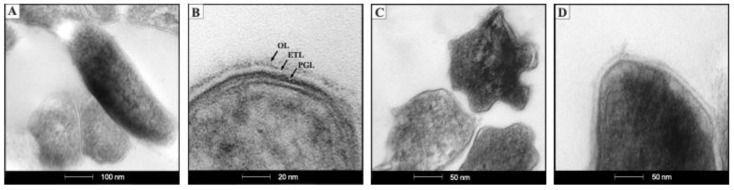
Representative ultrastructural micrographs of mycobacteria (upper row) and infected alveolar macrophages (lower row). (**A**) Typical morphology of *Mtb*H37Rv strain. (**B**) High power micrograph of the cell envelope of *Mtb*H37Rv shows three layers: an electron-dense outer layer (OL), electron transparent layer (ETL) and peptidoglycan layer (PGL). (**C**) Abnormal morphology of the *Mtb*∆*ctpF* strain, showing an irregular shape with projections and concavities on the bacterial surface. (**D**) High power micrograph of the cell envelope of *Mtb*∆*ctpF* showing thinner outer and peptidoglycan layers. (**E**) Normal ultrastructure of non-infected MH-S cells. (**F**) At 3 days post-infection (dpi) macrophages show a phagosome with electron-dense long rods that correspond to *Mtb*H37Rv (arrow). (**G**) At 3 dpi with *Mtb*∆*ctpF*, macrophages show numerous vacuoles with granular electron-dense material or small bacteria (arrow).

**Figure 3 ijms-23-06015-f003:**
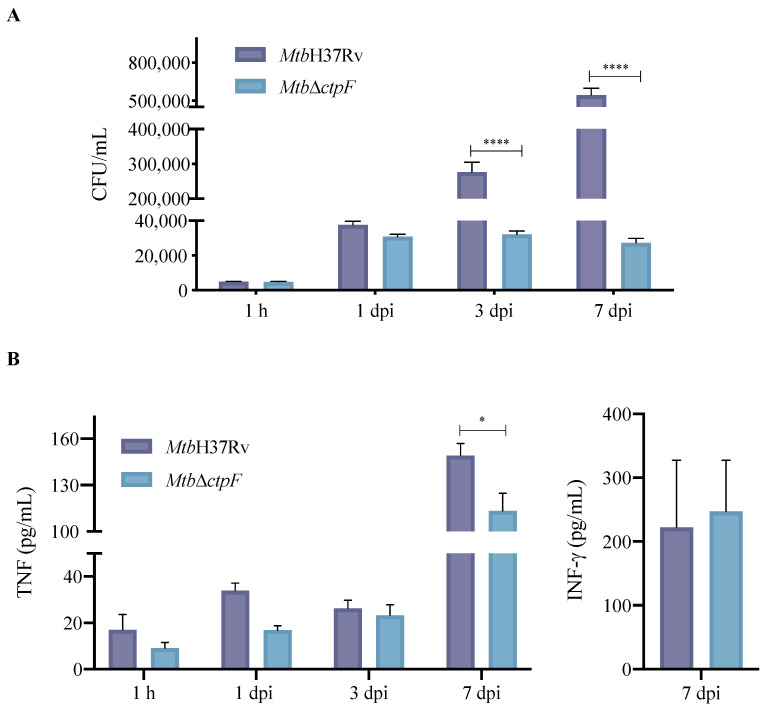
Proliferation of *Mtb*H37Rv and *Mtb*Δ*ctpF* strains in MH-S cells, and cytokine production in infected alveolar macrophages. (**A**) Infection assays were carried out with a multiplicity of infection (MOI) of 1:2 (1 × 10^5^ cells: 2 × 10^5^ bacteria in the logarithmic phase) at different infection times (1 h, and 1, 3 and 7 dpi). Bars represent CFU/mL ± standard error of the mean (SEM) derived from three parallel and independent infections. Data were analyzed using two-way analysis of variance (ANOVA) with Šídák’s multiple comparisons test (**** *p* < 0.0001). (**B**) The concentration of tumor necrosis factor (TNF) and interferon-gamma (IFN-γ) was determined in supernatants of MH-S in different infection time periods (1 h, 1, 3 and 7 dpi). Bars correspond to the average concentration of each cytokine (pg/mL) ± SEM derived from three parallel and independent infections. TNF and IFN-γ data were tested using Student’s t-comparison test (unpaired) (* *p* < 0.05).

**Figure 4 ijms-23-06015-f004:**
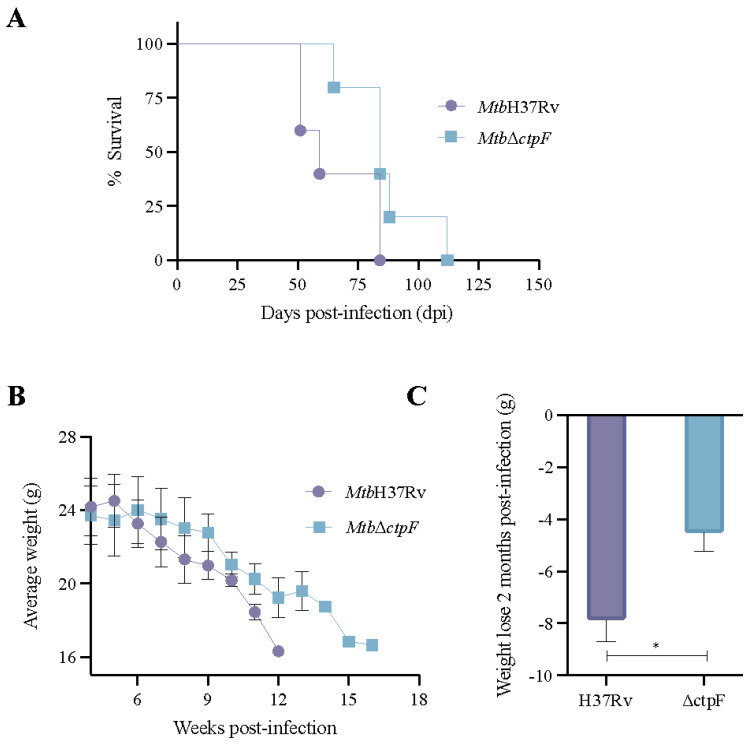
Survival of BALB/c mice infected with *Mtb*H37Rv and *Mtb*Δ*ctpF* strains. (**A**) Survival curves of BALB/c mice (5 animals per strain) inoculated intratracheally with 2.5 x10^5^ CFU/100 μL of *Mtb* strains. The curves were obtained by plotting the number of surviving mice as a function of time using the Kaplan–Meier algorithm. (**B**) The 5 mice groups were weighed weekly. Plotted values correspond to the average weight (g) ± SEM of animals in the same group. (**C**) Weight loss was calculated as the difference of the mean weight (g) ± SEM of the mice infected with parental and mutant strains at 4 and 12 weeks post-infection. Asterisks indicate statistical significance (* *p* < 0.05) determined by Student’s t-comparison test (unpaired).

**Figure 5 ijms-23-06015-f005:**
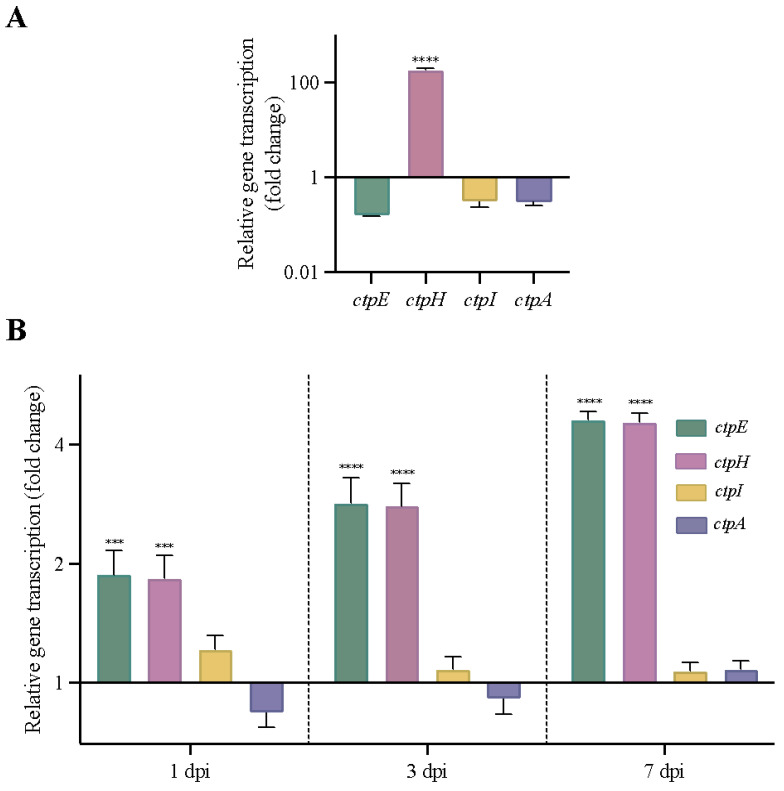
Transcriptional profile of genes encoding P2 type ATPases in *Mtb* strains. (**A**) Transcript levels are presented as the gene transcription ratio between cells intoxicated with 2.5 mM Ca^2+^ for 3 h and untreated *Mtb*Δ*ctpF* cells (control sample; transcription ratio ≈ 1.00). The plotted values correspond to the mean ± standard deviation of three technical replicates. (**B**) mRNA levels are expressed as the ratio between the number of cDNA copies of the mutant strain and the parental strain (control strain; transcription ratio ≈ 1.00) during the infection of MH-S cells. The plotted values correspond to the mean ± SEM of three parallel and independent in vitro infections. All the values were normalized to the level of 16S*rRNA* (*rrs*) mRNA, which was constant in both strains under all conditions studied. *ctpF* expression in *Mtb*Δ*ctpF* cells was not detected in either assay. An unrelated *ctpA* gene was included as a control. Asterisks indicate differences obtained by two-way ANOVA with Šidák’s multiple comparisons tests (*** *p* < 0.001 and **** *p* < 0.0001).

**Figure 6 ijms-23-06015-f006:**
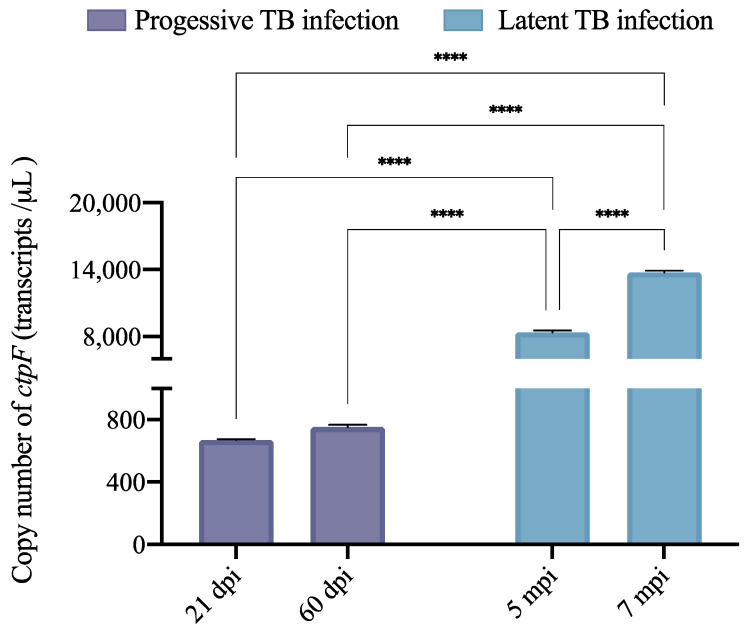
Comparison of the absolute expression (copy number/µL) of the *ctpF* gene in the lungs of mice with latent and progressive TB. The copy numbers of the *ctpF* gene in the lungs of three mice infected with *Mtb*H37Rv were determined by absolute qRT-PCR (3 mice per time point post-infection). The quantity of *ctpF* transcript was calculated according to a standard curve and the equation for linear regression. The data shown are the mean ± SEM of three different mice at each time point. The blue and purple bars represent the absolute expression of *ctpF* from animals with progressive or latent TB infection, respectively. Asterisks indicate differences obtained by one-way ANOVA with Šidák’s multiple comparisons test (**** *p* < 0.0001).

**Figure 7 ijms-23-06015-f007:**
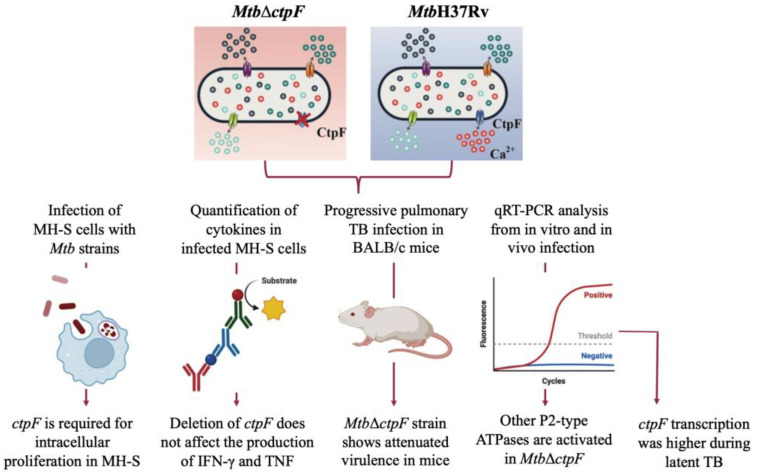
Scheme summarizing experimental procedures and the main results (Created with BioRender.com, accessed on 21 May 2022).

## Data Availability

The data presented in this study are available on request from the corresponding author.

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
