# Peer review of "The ctpF Gene Encoding a Calcium P-Type ATPase of the Plasma Membrane Contributes to Full Virulence of Mycobacterium tuberculosis"

_ijms, 2022, doi:10.3390/ijms23116015_

Round 1

Reviewer 1 Report

The manuscript by Maya-Hoyos et al. describe the influence of calcium ATPase on virulence of Mycobacterium tuberculosis. This work seems to be interesting for medicine investigations, however, I have some remarks.

  • The calcium transport across plasma membrane of Mycobacterium tuberculosis should be described in detail (including essential transporters, calcium ATPases, ion calcium channels).
  • The calcium signalling of Mycobacterium tuberculosis should be described in detail (including the role of calcium on growth, defence system and others).
  • The role of calcium ATPase for bacterial survival should be described in more detail.
  • Please, add scale in Figure 2. Additionally, add descriptions of arrows in Figure 2F and 2G.
  • The P2-type ATPases (ctpF, ctpH, ctpE, and ctpI) should be described in more detail. What are differences of these ATPase (direction of ion flux, number of ions and ATP per cycle and others)?
  • Please, summarize results of work in scheme. It can improve understanding of this work by potential readers.

Author Response

Review Report 1

Reviewer(s)' Comments to Author(s):

The manuscript by Maya-Hoyos et al. describe the influence of calcium ATPase on virulence of Mycobacterium tuberculosis. This work seems to be interesting for medicine investigations, however, I have some remarks.

Answer

We thank the referee for their positive comments.

Item 1

The calcium transport across plasma membrane of Mycobacterium tuberculosis should be described in detail (including essential transporters, calcium ATPases, ion calcium channels).

Answer

As the reviewer suggested, information about the calcium transport across plasma membranes of bacteria, including M. tuberculosis (Mtb), was included in the Introduction of the new version of the manuscript (Introduction, lines: 82-132; Discussion, lines: 350-367). Specifically, information about proteins containing Ca2+ binding domains (CaBPs) and the Mtb Proteinas PE_PEGRs, which contribute to regulating the Ca2+ content inside the mycobacteria, as well as helping to reduce metal concentrations in the phagosome, thus promoting the mycobacterial proliferation by hindering the phagolysosome fusion.

Item 2

The calcium signaling of Mycobacterium tuberculosis should be described in detail (including the role of calcium on growth, defense system and others).

Answer

As the reviewer suggested, information about the calcium signaling of Mtb was included in the new version of the manuscript (Introduction, lines: 77-81; Discussion, lines: 343-349; 387-392)

Item 3

The role of calcium ATPase for bacterial survival should be described in more detail

Answer

Thank you very much for this comment, information about the role of calcium ATPase for bacterial survival was included in the new version of the manuscript (Introduction, lines: 92-139)

Item 4

Please, add scale in Figure 2. Additionally, add descriptions of arrows in Figure 2F and 2G.

Answer

Thank you very much for highlighting this important wording detail, the scale and description of arrows were included in the Figure 2 caption (F and G; lines: 185-188), as follow:

Figure 2. (F) After 3 days post-infection (dpi) macrophages show a phagosome with electron-dense long rods that correspond to MtbH37Rv (arrow). (G) After 3 dpi with Mtb∆ctpF, macrophages show numerous vacuoles with granular electron-dense material or small bacteria (arrow).

Item 5

The P2-type ATPases (ctpF, ctpH, ctpE, and ctpI) should be described in more detail. What are differences of these ATPase (direction of ion flux, number of ions and ATP per cycle and others)?

Answer

Information about the specificity of ion transport and direction of ion flux mediated by CtpE, CtpF and CtpH were included in the new versión, where the introduction was rewritten (Introduction, lines: 109-132). At date, binding stoichiometry of CtpE, CtpF and CtpH have not yet been studied.

Item 6

Please, summarize results of work in scheme. It can improve understanding of this work by potential readers

Answer

A scheme summarizing the main results obtained has been included as a new figure as part of the conclusions section (Figure 7; lines: 662-666). 

Reviewer 2 Report

In this study Maya-Hoyos at el. studied the ΔctpF M. tuberculosis strain in contrast to MtbH37Rv. They found that the deletion of ctpF Ca2+ P-type ATPase does not affect the production of TNF and interferon-γ in infected cells. However, lack of ctpF was found to be responsible for Mtb proliferation in cells, and thus manifested in attenuated virulence in mice.  

This study is well written and the results are interesting, however the aim is not clearly communicated. The authors suggest that the current BCG vaccine should be replaced and their study could be relevant for an attenuated anti-TB vaccine development. However, their results show that MtbΔctpF is still lethal for mice, although it is less virulent than MtbH37Rv.

The authors should give more detail on the current vaccine and should explain why MtbΔctpF could be an attenuated vaccine candidate, if it still lethal.

Detailed comments are below:

  • Fig 3A: label should be CFU
  • Second graph of Fig 3B shows IFN-γ levels only after 7 days. It is not clear from the manuscript, if these levels were not checked before or if they were checked, but level was below detection limit. It should be included in the graph, even if levels are not detectable.
  • Fig 5A: It should be stated, how much time after Ca2+ treatment were these levels measured.
  • It should also be clearly stated how many animals were used for the distinct measurements.
  • Discussion only contains a summary of the results, that should rather be in the Results section.
  • Conclusion is fairly short and it does not interpret the results.

Author Response

Review Report 2

Reviewer(s)' Comments to Author(s):

In this study Maya-Hoyos at el. studied the ΔctpF M. tuberculosis strain in contrast to MtbH37Rv. They found that the deletion of ctpF Ca2+ P-type ATPase does not affect the production of TNF and interferon-γ in infected cells. However, lack of ctpF was found to be responsible for Mtb proliferation in cells, and thus manifested in attenuated virulence in mice.

Item 1

This study is well written and the results are interesting, however the aim is not clearly communicated. The authors suggest that the current BCG vaccine should be replaced and their study could be relevant for an attenuated anti-TB vaccine development. However, their results show that MtbΔctpF is still lethal for mice, although it is less virulent than MtbH37Rv.

Answer

We took this comment into account in our revised manuscript. It is important to consider that MtbΔctpF, (which was designed and constructed in this work), is only a mutant strain that cannot yet be considered as a vaccine candidate. At this point, we only know that deleting the ctpF gene produces impaired virulence in Mtb. If we are planning to perform preclinical assays using MtbΔctpF, we probably have to use a Mtb strain different than H37Rv, and to construct an unmarked double mutant also deleting another not correlated gene considered as a virulence factor of Mtb (Discussion, lines: 434-441).

Item 2

The authors should give more detail on the current vaccine and should explain why MtbΔctpF could be an attenuated vaccine candidate, if it still lethal.

Answer

The reviewer's comment is key for the future projection of the work described in this manuscript. Therefore, information about MtbΔctpF as a possible vaccine candidate is stated in the final paragraph of discussion (Discussion, lines: 434-441).

Item 3

Fig 3A: label should be CFU

Answer

We apologize for this oversight; the label was corrected in Fig. 3A (lines: 209-22). We were also corrected in the legend of Figure 4 (lines: 267-274).

Item 4

Second graph of Fig 3B shows IFN-γ levels only after 7 days. It is not clear from the manuscript, if these levels were not checked before or if they were checked, but level was below detection limit. It should be included in the graph, even if levels are not detectable.

Answer

As stated by the reviewer, the levels of IFN-γ were measured at different infection times (1 h, 1, 3 and 7 dpi), but they were only detectable at 7-day post-infection. This point was clarified in the result section and Figure 3B caption (Results, lines: 238-240 and legend of Figure 3, lines: 216-220)

Item 5

Fig 5A: It should be stated, how much time after Ca2+ treatment were these levels measured

Answer

Thank you for this comment, the period of time in which the Mtb wild type and mutant strain were 3 h; therefore this procedure detail was now included in the caption of Figure 5 (Results, lines 282-285; legend of figure 5, lines: 289-291)

Item 6

It should also be clearly stated how many animals were used for the distinct measurements.

Answer

As requested by the reviewer, the number of mice used in the progressive pulmonary and latent tuberculosis infection experiments are stated in material and methods (lines: 531-542; 568-577), Results (lines: 251-253), and legend of Figure 4 (lines: 267-274 and Figure 6 (lines: 329-336).

Item 7

Discussion only contains a summary of the results, that should rather be in the Results section.

Answer

Thank you very much for this suggestion that improves the understanding of the manuscript. As the reviewer suggested, part of the discussion text was removed and the discussion was rewritten (lines: 343-392; 434-441)

Item 8

Conclusion is fairly short and it does not interpret the results.

Answer

This is a pertinent observation; therefore, more information about results were included in the conclusion (lines 651-676) in order to improve the understanding of the whole manuscript.

Finally, we look forward to having our paper published in “International Journal of Molecular Sciences” and we extend our gratitude to the reviewers for their time and proper suggestions to make this manuscript stronger.

Round 2

Reviewer 1 Report

The manuscript was improved. I have not other questions.

Reviewer 2 Report

The manuscript has improved a lot and both the introduction and discussion read well.

Extensive English editing is needed. Some typos:
Line 73 ’ they preserve ion concentration at the suitable nutrient level for proper cellular function
Line 89 attenuate
Line 399 In addition
Line 664 Either